# Risk Assessment of Lack of Water Supply Using the Hydraulic Model of the Water Supply

**Krzysztof Boryczko \***, **Izabela Piegdoń, Dawid Szpak** and **Jakub Żywiec**

Department of Water Supply and Sewerage Systems, Faculty of Civil, Environmental Engineering and Architecture, Rzeszow University of Technology, Al. Powstancow Warszawy 6, 35-959 Rzeszow, Poland; piegi@prz.edu.pl (I.P.); dsz@prz.edu.pl (D.S.); j.zywiec@prz.edu.pl (J.Ż.)
\* Correspondence: kb@prz.edu.pl; Tel.: +48-17-865-1427

**Abstract:** Modern management of water supply systems is based on a preventive strategy consisting of the prevention of failures and crisis situations. Water pipe failures resulting in a lack of water supply for a long period pose a threat to the water consumers safety. The aim of the work was to present the methodology and develop a risk map of lack of water supply to consumers. The article presents a failures simulation of the main pipes transporting treated water from the water treatment plant to the city carried out using the EPANET 2.0. software. The simulation results made it possible to determine the consequences of failures by determining the number of inhabitants (consumers) affected with lack of water supply as a result of failure of the main pipes near the water treatment plant WTP which, together with the failure rate, were used to prepare risk maps of lack of water supply. The developed method was presented on the water supply network located in Central and Eastern Europe. It was found that the highest risk of lack of water supply is related to the failure of the M3 main pipe, which transports water to the eastern and north-eastern parts of the city. It is recommended to modernize the M3 main pipe, which will reduce the number of failures resulting in a lack of water supply.

**Keywords:** water supply network; failures; risk maps; Epanet

## 1. Introduction

The operation of water supply systems (WSS) is aimed at providing consumers with water of appropriate quality, in the right amount and under the appropriate pressure [1–3]. Increasingly, the activity of water supply companies is based on a preventive strategy consisting of the prevention of failures and crisis situations. The basis of this strategy is a risk assessment based on the detailed identification of hazards and the validation of existing safety measures. This is consistent with the World Health Organization (WHO) guidelines on Water Safety Plans (WSP) [4,5]. Reducing the number of undesirable events increases the safety of water supply to consumers [1,3,6]. Various IT tools are used to support the work of water companies. IT tools for managing WSS can be divided into four groups: Geographic Information System (GIS), Supervisory Control And Data Acquisition (SCADA), Enterprise Resource Planning (ERP) and hydraulic models [7–23]. The integrated operation of these tools allows detailed information to be obtained that can be used for monitoring the work in real mode and and to archive the obtained data and their subsequent use. Managing the WSS, implemented by an integrated IT system is aimed, inter alia, at visualization of the WSS, determination of the current operating status, optimization and design of the WSS or the location of failures and hidden water leaks [7,9,10,15]. The risk assessment in the WSS can be performed based on data obtained from IT tools, especially from the hydraulic model, which will allow for supplementing and extending the risk matrix analysis method recommended for WSP [10,11,13,22,24–26].

One of the main difficulties in implementing WSP signaled by the industry is the inability to define risk and the uncertainty in assigning point weights to individual risk

factors. The World Health Organization recommends, among others, using a two-parameter risk matrix that presents risk as the product of the probability of occurrence of undesirable events and losses resulting from their occurrence [4]. It seems that this is the correct approach. However, one should refrain from adopting the point weights of the input parameters subjectively without reference to information obtained from the monitoring system or simulations. Comprehensive risk assessment related to the functioning of the WSS should take into account both the events related to the lack of water supply and water pollution. The developed method focuses on the amount of water. The safety theory does not cover all technical failures, but only those that may pose a threat to human health and life. For this reason, the work focuses on failures of the main pipes near the water treatment plant (WTP), which may result in a long lack of water supply to a large part of consumers. Due to high pressure, large diameters, and hence, a large amount of water flowing out of the pipe during a failure, their effects are significant and difficult to remove. The developed method is based on data on the failures of the WSS and a hydraulic model (Epanet 2.0) that works on the basis of historical data. The Epanet (United States Environmental Protection Agency, Washington D.C., USA) software simulated the effects of failures on the main pipes which was expressed by the number of inhabitants who do not have water as a result of the pressure drop.

Risk maps are widely used in various fields and there is a lot of research on this topic. It is a simple and effective tool that can be used to quickly identify high-risk areas. Most often, the probability of undesirable events and its consequences are taken into account. In the broadly-understood water management, the most common are flood risk maps, which are used in many countries as one of the tools for environmental decisions. An extensive, multi-criteria approach to the assessment of flood risk and the creation of flood risk maps is presented in [27]. Risk maps in WSS are not widely used, so there is not much research on this topic. This is due to the fact that the WSS management based on IT tools is a relatively new approach, especially in developing countries. The maps can relates risk associated with quantity of supplied water and risk associated with quality of supplied water [28]. Until now, research on risk maps associated with quality of supplied water have been performed mainly in developing countries, where contaminated water causes many serious diseases. In work [24], mapping was performed and concentration of fluoride in drinking water was analyzed based on the GIS system. The exposure to trace metal contamination of drinking water sources in Pakistan was analyzed in the study [29]. Analysis associated with quantity of supplied water are often based on water pipe failures simulation in a hydraulic model and pressure drop analysis [15,22,25,26,28,30]. The aim of the work was to present the methodology and to develop a risk map of lack of water supply to consumers for a selected city in Central and Eastern Europe. The article presents a failures simulation of the main pipes transporting treated water from the WTP to the city using the Epanet 2.0 program. Based on the number of inhabitants (consumers) affected with lack of water supply as a result of failure of the main pipes near the WTP and the failure rate values for these pipes the risk of lack of water supply has been determined. On this basis a risk map was developed taking into account the three-level risk scale, i.e., low, medium and high.

## 2. Materials and Methods

### 2.1. Characteristics of the Research Object

The research object is the water supply network (WSN) located in south-eastern Poland. It supplies approximately 200,000 inhabitants with water and covers an area of 120 km$^2$. The area is administratively divided into 30 areas as shown in Figure 1.

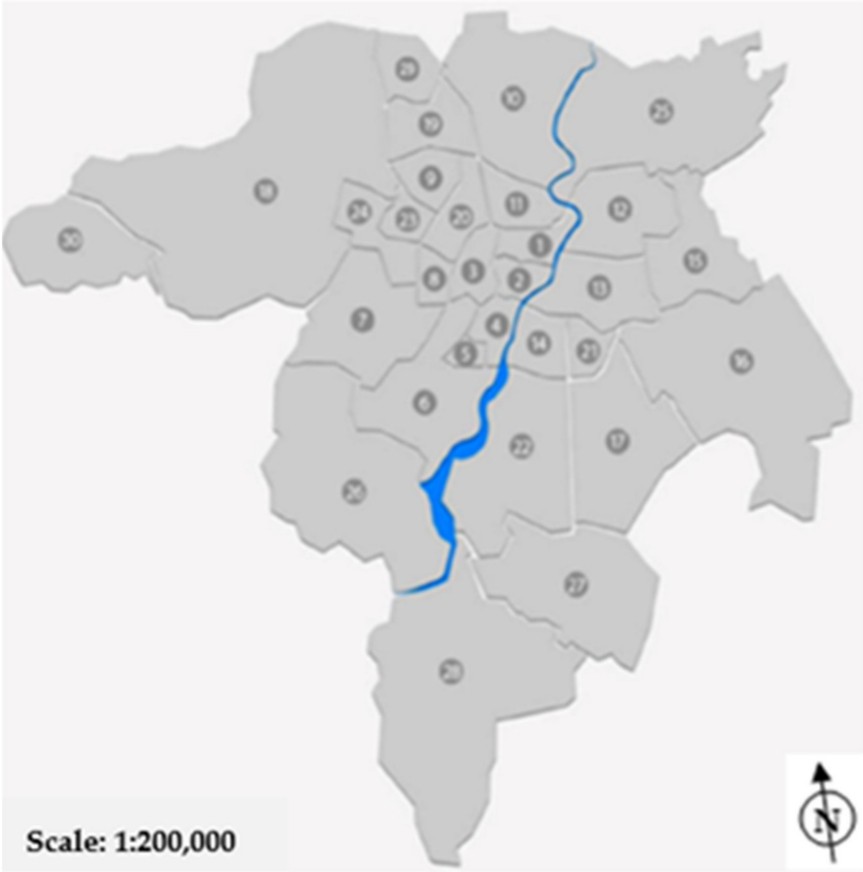

Scale: 1:200,000

**Figure 1.** Administrative division of the city area covered by the analysis.

The total length of the WSN is 1025 km, of which 55 km is the main network, 605 km is the distribution network and 365 km are house connections. The system has been in use for 80 years, of which approximately 35% of pipes have been in use for less than 10 years, approximately 23% of pipes have been in use for 11–25 years, approximately 30% of pipes have been in use for 26–50 years, the oldest pipes have been in use for longer than 50 years make up about 12% of all causes. 43% of the network is made of PE, 32% are PVC pipes, 17% are cast iron pipes, 4% of pipes are made of steel and 4% of other materials. The network operates at 80% in a closed system. The network cooperates with two groups of water tanks: ZB1 and ZB2, which are located in the eastern and western parts of the city, with a total capacity of about 35,000 m$^3$. The scheme of the WSN has been shown in Figure 2. A skeleton of the WSN are 4 main pipes transporting treated water from the WTP to the city:

- M0 main pipe—supplies the north western part of the city and tanks ZB1,
- M1 main pipe—transports water to the central and northern parts of the city,
- M2 main pipe—transports water to the southern and central parts of the city,
- M3 main pipe—transports water from WTP to the eastern and north- -eastern parts of the city, supplying tanks ZB2.

The initial sections of these pipes, which are not connected to each other, play a key role. For this reason, a failure rate analysis was carried out for the initial sections of main pipes, which was the basis for the risk assessment of the lack of water supply to consumers in individual administrative areas. Table 1 presents the data on the initial main pipes sections. In Figure 2, the colors indicate the location of the initial main pipes sections: M0—green line, the M1—blue line, M2—red line, M3—pink line.

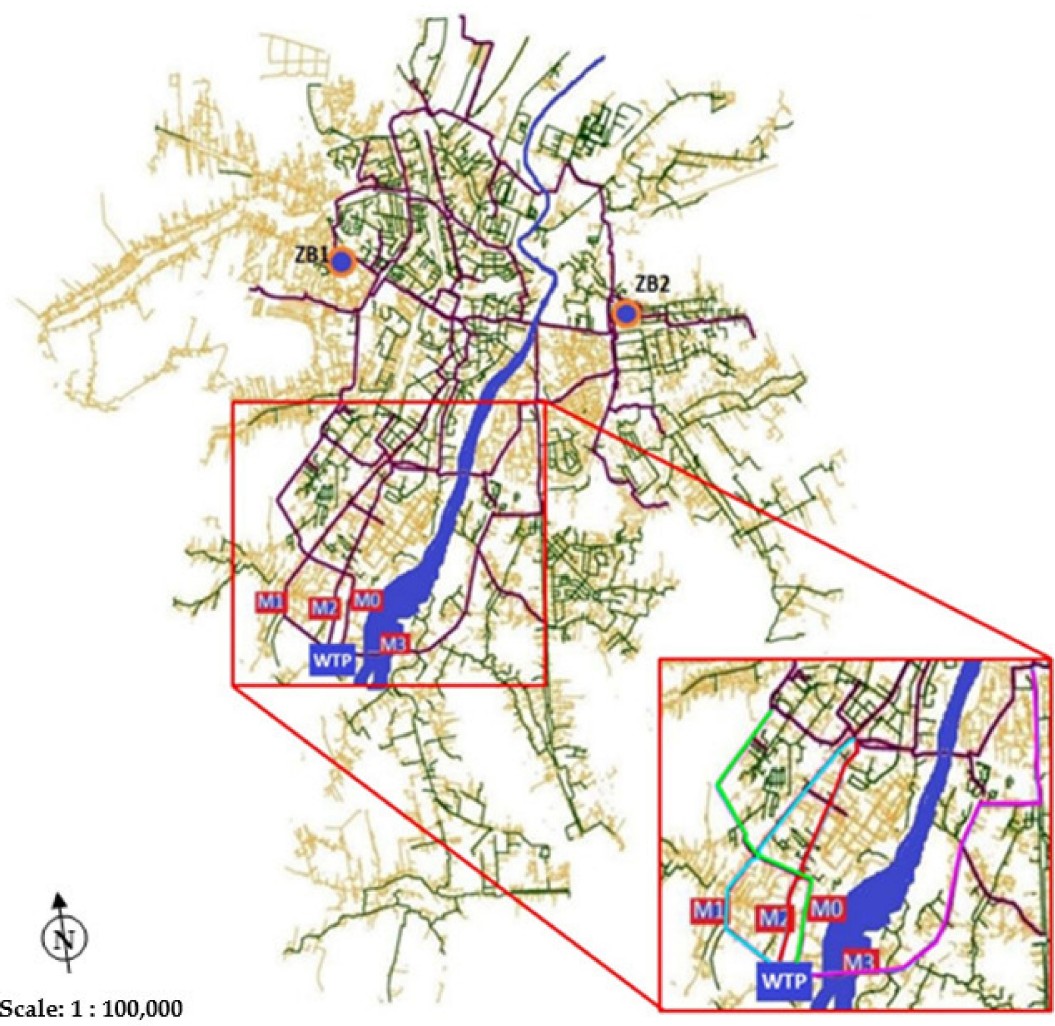

**Figure 2.** The scheme of the water supply network.

**Table 1.** Information on main pipe sections accepted for analysis.

| Main Pipe | Diameter | Length | Material |
|:---:|:---:|:---:|:---:|
| | mm | km | |
| M0 | 1200 | 3.92 | gray cast iron |
| M1 | 450 | 3.86 | gray cast iron |
| M2 | 400 | 3.04 | PE |
| M3 | 400 | 5.76 | gray cast iron |

## 2.2. The Risk Matrix for Lack of Water Supply

The failure simulation was made on the basis of the Authors' hydraulic model of the WSN [26]. The model has existed for several years and is constantly updated with new pipes and objects. The model was developed in the EPANET 2.0. software and has all the input parameters that allow for hydraulic modeling of the WSN:

- pipe lengths are scaled according to the real map,
- terrain ordinates and pipe diameters are consistent with the map,
- water partitioning in nodes was assumed according to the number of inhabitants,
- characteristics of pumps and tanks were adopted on the basis of information provided by the water company.

In environmental engineering (as in most engineering applications) the basic risk definition applies, which presents risk as a product of the probability of undesirable events

occurrence and losses resulting from it. The paper proposes the use of the two-parameter risk matrix:

$$r_{ij} = P_i \cdot C_j \qquad (1)$$

In the Equation (1) the "$P_i$" stands for point weight related to the probability of undesirable events occurrence ($i$ = 1, 2, . . . , n, where "n" is the number of the scale adopted for the probability parameter); the "$C_j$" stands for point weight related to the consequences (lack of water supply) related to the occurrence of undesirable event ($j$ = 1, 2, . . . , m where "m" is the number of the scale adopted for the consequences parameter).

This approach is consistent with the WHO guidelines on WSP. According to the WHO guidelines [4], the risk associated with each hazard can be described by the probability of its occurrence (e.g., "low", "medium", "high") and the consequences of the hazard (e.g., "low", "medium", "high"). The potential impact on human health should be the most important element in the risk analysis, but other factors, such as continuity of water supply should also be taken into account. The water company should define in detail its understanding of the terms that are used. Risk factor point weights are often adopted on the basis of experience, knowledge and opinion of water company employees, good practices and technical literature. In order to avoid subjectivization of the risk assessment, one should strive to use the largest possible amount of numerical data that can be obtained from IT tools commonly used to manage WSS. The central point of the system is the GIS numerical map. Its use enables computer visualization of the WSN and precise location of failures on the map. The GIS system also enables the presentation of information from the monitoring system and the presentation of the results of hydraulic simulations.

It is proposed to adopt the values of P and C parameters on the basis of operational data obtained from the computer system for supervision over the water supply network, i.e., the results of the main pipes failure simulations (parameter C) and the failure rate determined on the basis of failure data (parameter P). The evaluation criteria for individual point weights are presented in Tables 2 and 3. The risk determined in accordance with (1) takes the values from the range: 1 to 25. The risk matrix is presented in Table 4. A standard three scale of risk levels is proposed:

- tolerated risk − 1 ÷ 3,
- controlled risk − 4 ÷ 12,
- unacceptable risk − 15 ÷ 25.

**Table 2.** Criteria and point weights for parameter P.

| Probability | Failure Rate $\lambda$ [Failure $\cdot$ km$^{-1}$ $\cdot$ year $^{-1}$] | P |
|---|---|---|
| very low | <0.3 | 1 |
| low | 0.3–0.5 | 2 |
| medium | 0.5–0.75 | 3 |
| high | 0.75–1.0 | 4 |
| very high | >1.0 | 5 |

**Table 3.** Criteria and point weights for parameter C.

| Consequences | Population without Water | C |
|---|---|---|
| very low | <100 | 1 |
| low | 100–500 | 2 |
| medium | 500–2000 | 3 |
| high | 2000–5000 | 4 |
| very high | >5000 | 5 |

**Table 4.** Risk matrix.

| P | C | | | | |
|---|---|---|---|---|---|
| | **1** | **2** | **3** | **4** | **5** |
| **1** | 1 | 2 | 3 | 4 | 5 |
| **2** | 2 | 4 | 6 | 8 | 10 |
| **3** | 3 | 6 | 9 | 12 | 15 |
| **4** | 4 | 8 | 12 | 16 | 20 |
| **5** | 5 | 10 | 15 | 20 | 25 |

In order to determine the value of the consequences parameter (parameter C), the failure rate was determined, which takes into account the number of failures and the length of the analyzed main pipes [13,30]:

$$\lambda = \frac{n(\Delta t)}{L \cdot \Delta t} \left[ failure \cdot km^{-1} \cdot year^{-1} \right] \tag{2}$$

In the Equation (2) the "$n(\Delta t)$" stands for the number of failures in the time interval $\Delta t$; the "$L$" stands for the length [km] of examined pipes in the time interval $\Delta t$ and the "$\Delta t$" is the considered period of time in years.

Due to the greatest consequences for water consumers, only the failures at the initial sections of main pipes that are most severe for water consumers are considered. The range of the analyzed main pipes is shown in Figure 2.

## 3. Results

### 3.1. Water Main Pipes Failure Simulation

For the analysis, the existing hydraulic model water distribution system WDS [26] was used, which was extended with water supply pipes for newly built housing estates, and was verified and updated with model input data, e.g., water consumption, length and diameter of pipes, roughness of pipes, volume of tanks, operating parameters of the pumping station. The developed model was calibrated by comparing the pressure measurement results from several measurement points on the main pipes of the water supply network with the results of the analysis using the Epanet software. The maximum difference in the pressure values did not exceed 1.5 m. As standard, pressure measurements on the analyzed water supply network at measuring points are carried out for external hydrants for the needs of the National Fire Department.

The work shows the simulation of the failure of the initial sections of the M0, M1, M2 and M3 pipes (complete closure of the flow for a period of 24 h) carried out using the EPANET 2.0. software. These pipes were selected because of the biggest consequences of their failure. Figure 3 shows the pressure distribution in the water supply network during normal failure-free operation. In the areas of low pressure value (i.e., areas <10 m in Figure 3) in the northern and eastern part of the city are located tall buildings powered by a hydrophore. Therefore, it does not affect the continuity of water supply. The water company is in charge of 39 local hydrophore units. From these facilities, 31 are currently in use, while 8 are out of service. These objects were not included in the model, hence, the red areas in Figure 3. The developed model is the first in the history hydrualic model for the analyzed city water supply network and includes the main pumping station and the most important pipes. It is sufficient to carry out a reliable simulation of the effects of main pipes failure and to present the developed methodology. Currently the detailed hydraulic model integrated with other IT tools used in the enterprise (GIS and SCADA systems), which can be used in real conditions and run a simulation based on historical data is being developed. The real-time model should be understood as a mathematical model powered by a SCADA system, which uses the recorded values as boundary conditions for calculations. The model will binclude all main and distribution pipes and for household connections greater than

or equal to DN80. It will be one of the main elements of the supervision computer system of the water supply network.

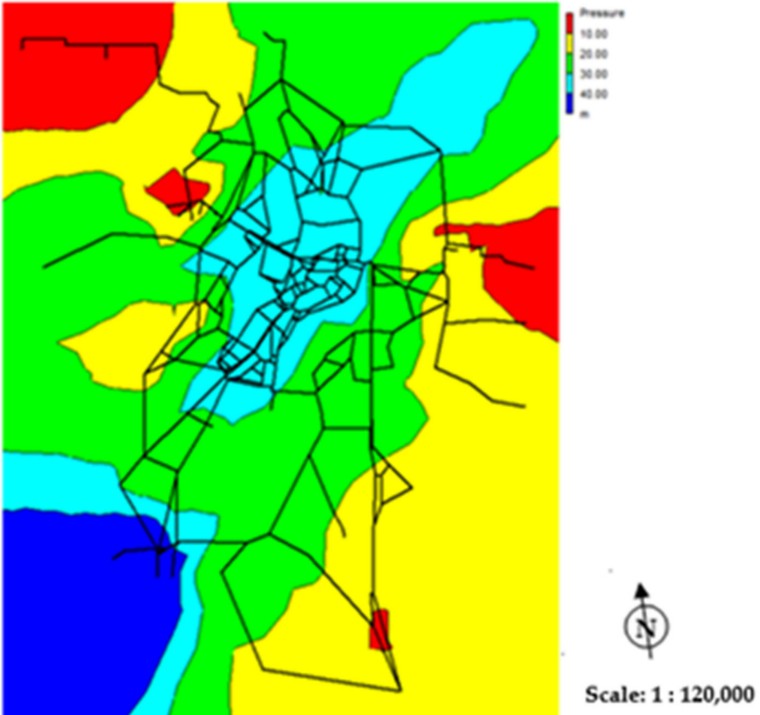

**Figure 3.** Pressure distribution during normal conditions.

The simulation results of the failure of four main pipes are shown in Figure 4. A decrease in pressure value below the required level causes limitation or suspension of water supply (no possibility of water consumption). The performed simulation did not show a pressure value decrease below the required pressure level in the central and northern parts of the city. This is due to the fact that the analyzed network was built as the looped network, and the city is supplied by four main pipes. Switching off 1 of 4 pipes does not cause a significant pressure value decrease and is not noticeable for most of the consumers. The inhabitants of areas located in the southern and eastern part of the city are the most exposed to risk of lack of water supply.

### 3.2. Failure Analysis of Water Main Pipes

Table 5 presents the values of failure rate, calculated with formula (1) for the main pipes M0, M1, M2 and M3 in the years 2004 ÷ 2018 and their basic statistics. The results of the analysis are presented graphically in Figure 5.

**Table 5.** Statistics of the main pipes M0–M3 failure rates.

| Statistics | Failure Rate $\lambda$ [1·km$^{-1}$·year$^{-1}$] | | | |
|---|---|---|---|---|
| | M0 | M1 | M2 | M3 |
| average | 0.02 | 0.93 | 1.38 | 1.32 |
| median | 0 | 0.78 | 1.32 | 1.22 |
| minimum | 0 | 0.26 | 0.33 | 0.87 |
| maximum | 0.26 | 1.81 | 3.62 | 2.43 |
| standard deviation | 0.07 | 0.42 | 0.98 | 0.47 |
| lower quartile (25%) | 0 | 0.52 | 0.98 | 0.87 |
| upper quartile (75%) | 0 | 1.17 | 1.97 | 1.56 |

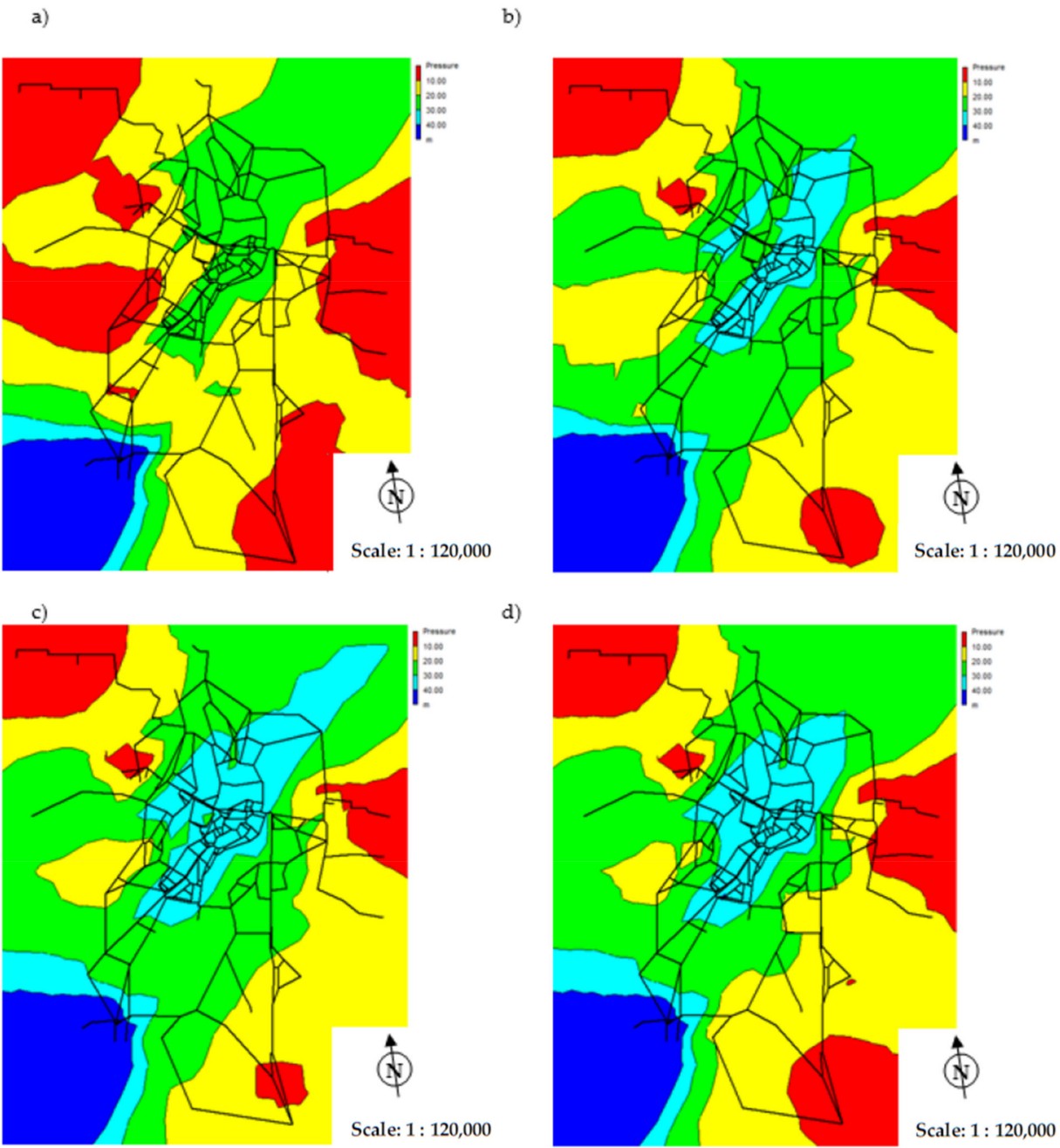

**Figure 4.** Pressure distribution during failure of the main pipes: (**a**) M0 pipe; (**b**) M1 pipe; (**c**) M2 pipe; (**d**) M3 pipe.

In the analyzed period, 232 main pipes failures occurred, which is: about 0.5% on the M0 pipe, about 23.5% on the M1 pipe, about 27% on the M2 pipe, and about 49% on the M3 pipe of all examined failures. The most common cause of water network failure is unsealing in gray cast iron pipes. There was only one failure on the M0 pipe in the tested period of 15 years, which proves it is in very good technical condition. The highest unit failure rate was observed for the M2 pipe, i.e., $\lambda_{avg}$ = 1.38 fail./km·year and for the M3 pipe i.e., $\lambda_{avg}$ = 1.32 fail./km·year. The average value of the failure rate determined for the M1 pipe is $\lambda_{avg}$ = 0.93 fail./km·year. The determined $\lambda$ values for M1, M2 and M3 pipes differ from the generally accepted limit value, which for the main network $\lambda$ = 0.3 fail./km·year. Failure of the initial sections of main pipes, due to the high values of the failure rates in the analyzed city is a significant source of risk of the lack of water supply, thus, the adopted assumption to simulate failures of these pipes is correct.

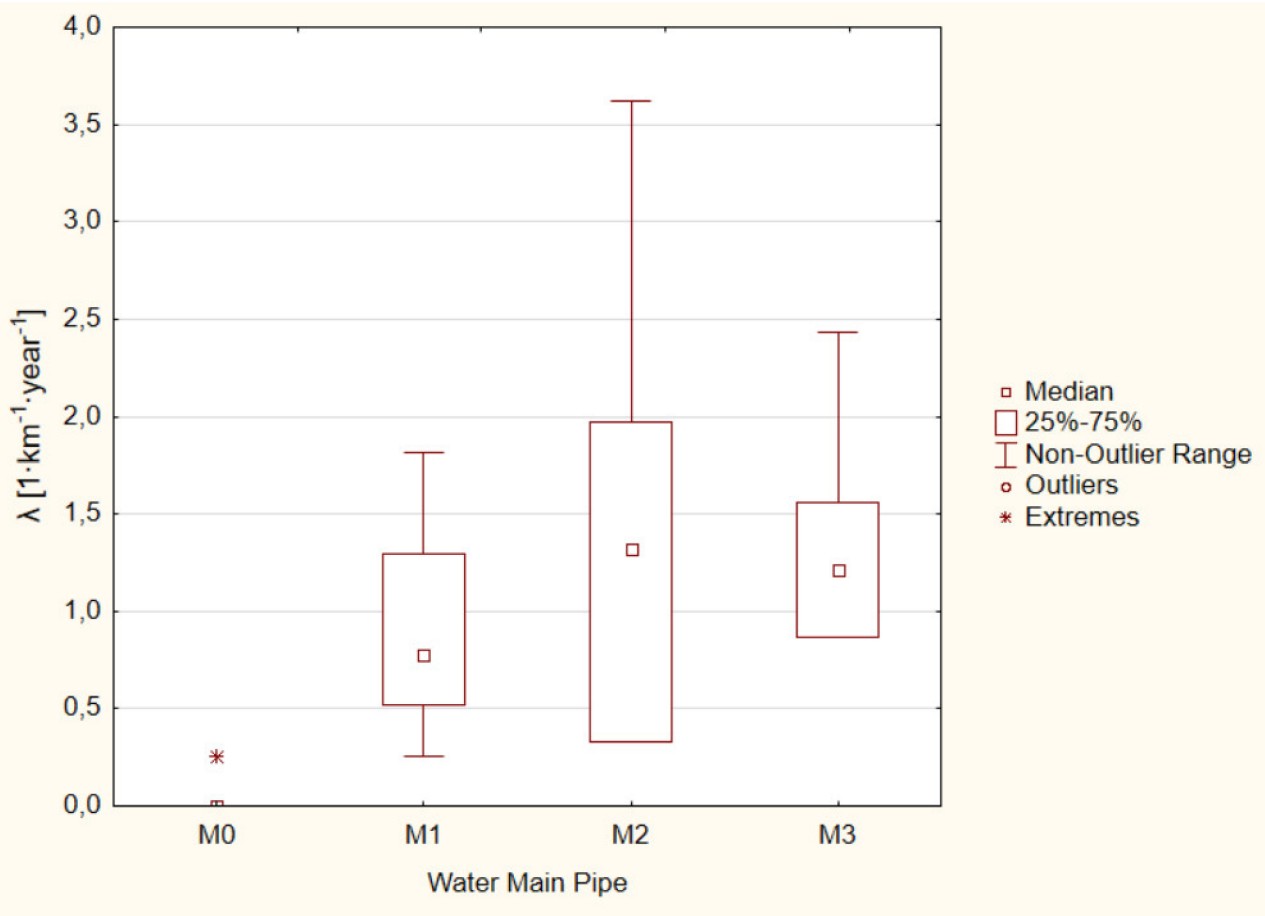

**Figure 5.** Failure rates of the M0, M1, M2 and M3 pipes.

### 3.3. The Risk of Lack of Water Supply to the Consumers

In accordance with the proposed methodology for determining the risk of the lack of water supply in a crisis situation, presented in point 2 of this study, the value of the risk of the lack of water supply was estimated for each of the city's residential areas. The failure of the M0, M1, M2 and M3 main pipes near the WTP was selected for the quantitative analysis of the risk related to the lack of water supply. The scale for the P parameters was adopted in accordance with Table 1. Probability of failure of the:

- M0 pipe: $\lambda_{avg}$ = 0.02 fail./km·year → P = 1 (very small),
- M1 pipe: $\lambda_{avg}$ = 0.93 fail./km·year → P = 4 (high),
- M2 pipe: $\lambda_{avg}$ = 1.38 fail./km·year → P = 5 (very high),
- M3 pipe: $\lambda_{avg}$ = 1.32 fail./km·year → P = 5 (very high).

The pressure distribution in water supply network obtained as a result of the simulation of the failure of the M0, M1, M2 and M3 main pipes (shutdown of the flow for a 24 h) in the area near to the WTP carried out in the EPANET 2.0 program are presented in Section 3.1. On the basis of the pressure in the network and the height of the buildings, the number of inhabitants affected with the lack of water supply was determined, as shown in Table 6. For parameter C (number of consumers affected with lack of water supply), point weights were adopted in accordance with Table 2. The risk value was calculated based on the formula (1). The water supply network supplying the area was divided into areas according to the city's administrative division.

Table 6. Number of inhabitants (consumers) affected with lack of water supply as a result of failure of the M0, M1, M2 and M3 main pipes near the WTP.

| Residential Area | The Number of Inhabitants (Consumers) | Number of Inhabitants Affected with Lack of Water Supply —Failure of M0 Pipe | Parameter C Point Weight—Failure of M0 Pipe | Number of Inhabitants Affected with Lack of Water Supply—Failure of M1 Pipe | Parameter C Point Weight—Failure of M1 Pipe | Number of Inhabitants Affected with Lack of Water Supply—Failure of M2 Pipe | Parameter C Point Weight—Failure of M2 Pipe | Number of Inhabitants Affected with Lack of Water Supply—Failure of M3 Pipe | Parameter C Point Weight—Failure of M3 Pipe |
|---|---|---|---|---|---|---|---|---|---|
| 1 | 4377 | 0 | 1 | 0 | 1 | 0 | 1 | 0 | 1 |
| 2 | 3358 | 0 | 1 | 0 | 1 | 0 | 1 | 0 | 1 |
| 3 | 4897 | 0 | 1 | 0 | 1 | 0 | 1 | 0 | 1 |
| 4 | 7530 | 0 | 1 | 0 | 1 | 0 | 1 | 0 | 1 |
| 5 | 4524 | 0 | 1 | 0 | 1 | 0 | 1 | 0 | 1 |
| 6 | 8604 | 0 | 1 | 0 | 1 | 0 | 1 | 0 | 1 |
| 7 | 3262 | 320 | 2 | 300 | 2 | 0 | 1 | 0 | 1 |
| 8 | 6876 | 0 | 1 | 0 | 1 | 0 | 1 | 0 | 1 |
| 9 | 11,322 | 0 | 1 | 0 | 1 | 0 | 1 | 0 | 1 |
| 10 | 4850 | 0 | 1 | 0 | 1 | 0 | 1 | 0 | 1 |
| 11 | 8791 | 0 | 1 | 0 | 1 | 0 | 1 | 0 | 1 |
| 12 | 3986 | 0 | 1 | 0 | 1 | 0 | 1 | 0 | 1 |
| 13 | 10,344 | 0 | 1 | 0 | 1 | 0 | 1 | 0 | 1 |
| 14 | 13,910 | 0 | 1 | 0 | 1 | 0 | 1 | 0 | 1 |
| 15 | 6106 | 0 | 1 | 0 | 1 | 0 | 1 | 820 | 3 |
| 16 | 7059 | 0 | 1 | 0 | 1 | 0 | 1 | 560 | 3 |
| 17 | 8350 | 0 | 1 | 0 | 1 | 0 | 1 | 840 | 3 |
| 18 | 9685 | 0 | 1 | 0 | 1 | 0 | 1 | 0 | 1 |
| 19 | 10,622 | 0 | 1 | 0 | 1 | 0 | 1 | 0 | 1 |
| 20 | 5318 | 0 | 1 | 0 | 1 | 0 | 1 | 0 | 1 |
| 21 | 4167 | 0 | 1 | 0 | 1 | 0 | 1 | 400 | 2 |
| 22 | 12,706 | 0 | 1 | 0 | 1 | 0 | 1 | 0 | 1 |
| 23 | 7716 | 0 | 1 | 0 | 1 | 0 | 1 | 0 | 1 |
| 24 | 5718 | 0 | 1 | 0 | 1 | 0 | 1 | 0 | 1 |
| 25 | 2323 | 0 | 1 | 0 | 1 | 0 | 1 | 0 | 1 |
| 26 | 3529 | 1060 | 3 | 700 | 3 | 0 | 1 | 0 | 1 |
| 27 | 2549 | 760 | 3 | 260 | 2 | 130 | 2 | 1000 | 4 |
| 28 | 959 | 480 | 2 | 100 | 2 | 60 | 1 | 450 | 2 |
| 29 | 6008 | 0 | 1 | 0 | 1 | 0 | 1 | 0 | 1 |
| 30 | 626 | 220 | 2 | 0 | 1 | 0 | 1 | 0 | 1 |

The map of risk of the lack of water supply for the four considered cases (failure of the M0, M1, M2 and M3 pipes) is presented in Figure 4.

Based on Figure 6, it was found that the area 27 is the most exposed to the risk of a lack of water supply. This area consist of single-family houses, small service buildings and schools. The highest risk of lack of water supply is related to the failure of the M3 main pipe, which transports water to the eastern and north-eastern parts of the city and supplies the ZB2 reservoirs. This is due to the fact that the M3 main pipe has a high failure rate (it is made of gray cast iron) and it is the only pipe supplying the eastern part of the city for which a high risk of lack of water supply was obtained. The water supply company should implement appropriate measures to reduce the risk. Due to the inconvenience for water consumers, modernization of the M3 main pipe should be considered. The analyzed fragment of the M3 pipe is 5.76 km long, which is about 0.5% of the total length of the network. Therefore, it seems that such action should be feasible for the water supply company. However, this decision should be preceded by an assessment of the company's finances and the nuisance of the renovation work for people and the environment. It is recommended that trenchless technology is used. The risk of lack of water supply in the case of a failure of the M0 main pipe is small. The recommended action is to monitor the existing pipe condition, flow rate and pressure measurement and monitoring, which will allow for quick detection of emergency situations. The average risk of a lack of water supply was obtained in the case of simulating the failure of M1 and M2 pipes. Despite the high failure rate of these pipes, which proves that they are not in the best technical condition, their location does not pose a high risk of lack of water supply. However, it should be expected that their technical condition will deteriorate and their modernization should be considered in the longer term.

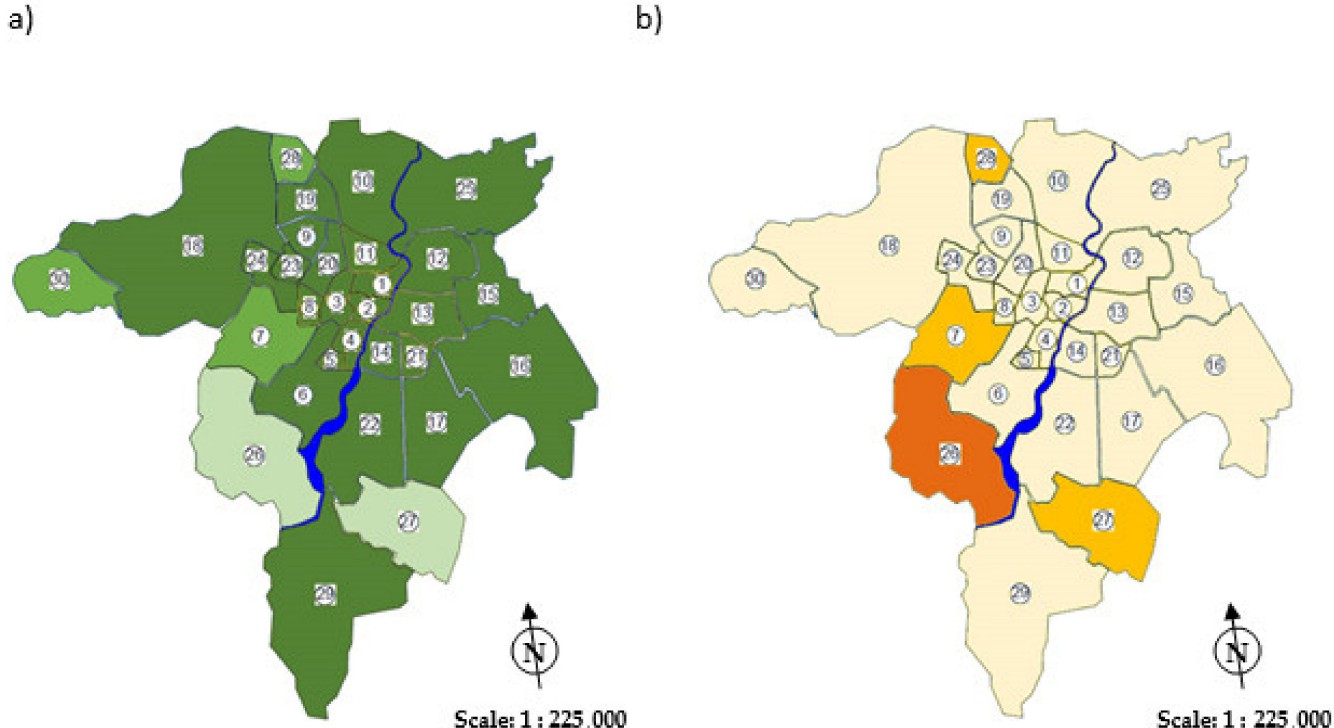

**Figure 6.** *Cont.*

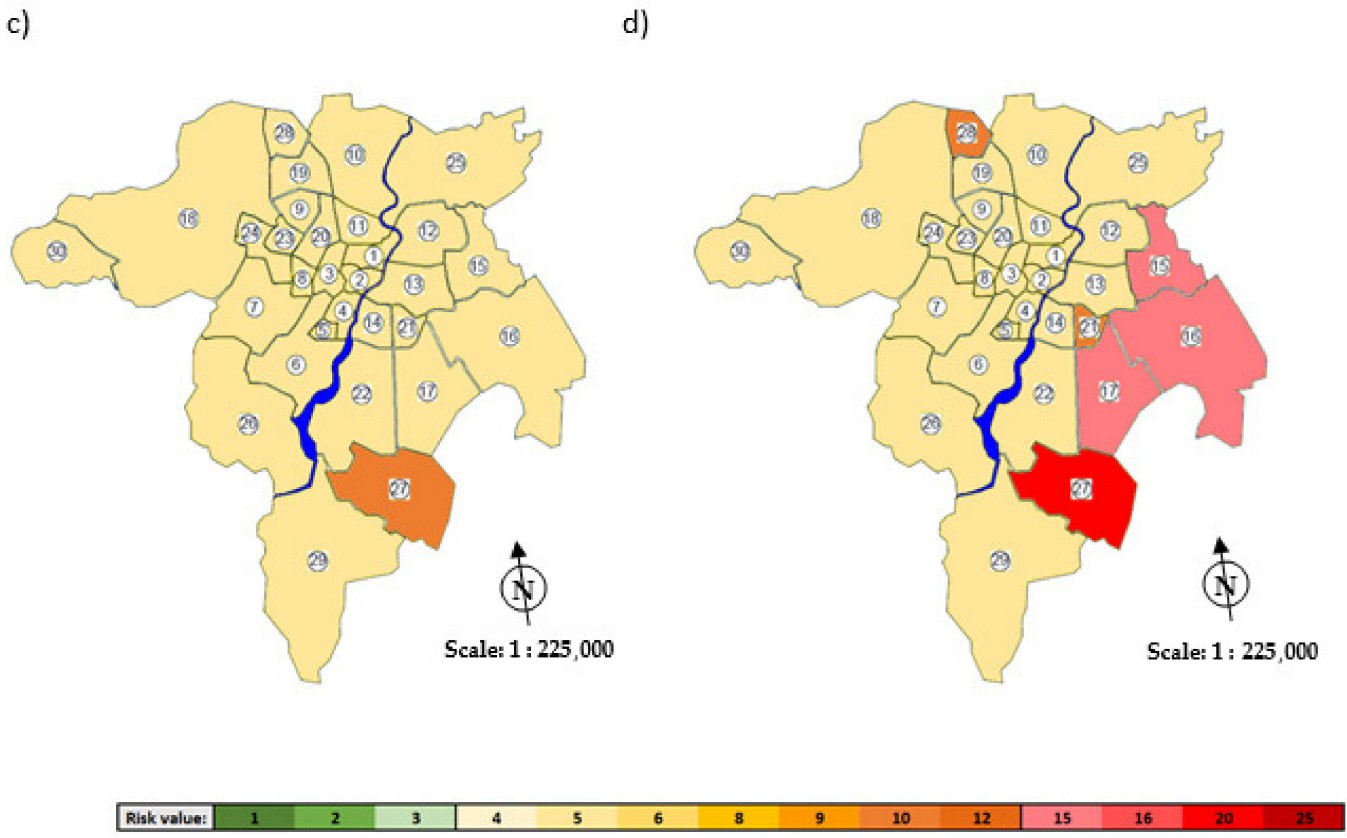

**Figure 6.** Map of risk of water supply lack in case of failure of the: (**a**) M0 main pipe; (**b**) M1 main pipe; (**c**) M2 main pipe; (**d**) M3 main pipe.

## 4. Conclusions and Perspectives

It is possible to simulate the closure (i.e., failure) of individual pipes in the water supply network, and then a comparative analysis of the pressure distribution in this network in failure-free conditions and during failure. In this way, it is possible to determine the consequence of failures on individual pipes in relation to the area where the pressure value dropped below the required level, the duration of this pressure reduction and the potential number of consumers affected by the limitation of water supply.

The presented method is based on failure-data of the water supply network and a simulating hydraulic model. The usefulness of the method depends on the reliability of the input data. The proposed approach is in line with the WHO guidelines on risk analysis in water supply systems.

This study focuses on the quantitative aspect of drinking water supply, the comprehensive analysis should also take into account the qualitative aspect, which can also be done in EPANET software (e.g., EPANET-MSX) [16,23,24,29]. Based on the results obtained, strategies can be developed to prevent main pipes failures that cause long-term water supply outages. The simulation results indicated that the highest risk was posed by the failure of the M3 main pipe, which led to the conclusion that it requires renovation. The analyzed network was made as a looped network, and the city is supplied from four main pipes, thus, for the most of housing estates, there is no high risk of lack of the water supply, even during a long-term failure of the city's main water pipes.

The method is fully reproducible and can be used to analyze the operation of other water supply networks. It can be especially useful for water supply networks with one intake or a small number of pipes supplying the city, i.e., small water supply systems. The method can be the basis of an operational strategy aimed at maintaining water supplies, which are now becoming reliability-oriented, i.e., monitoring, surveys and preventive maintenance.

**Author Contributions:** All authors equally contributed to the development of this manuscript. All authors have read and agreed to the published version of the manuscript.

**Funding:** This research received no external funding.

**Acknowledgments:** We thank the reviewers for their feedback, which helped to improve the quality of the manuscript.

**Conflicts of Interest:** The authors declare no conflict of interest.

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
