# Peer review of "Risk Assessment of Lack of Water Supply Using the Hydraulic Model of the Water Supply"

_resources, doi:10.3390/resources10050043_

Round 1
Reviewer 1 Report
The paper is interesting and clearly described. The only scientifically important aspect, not addressed by the Authors, is the use of Epanet 2.0 with a Demand Driven (DD) approach rather than the use of Epanet 2.2 with a Pressure Driven (PD) approach. The PD approach, when simulating breakages or inefficiencies of a water distribution system, certainly provides results closer to reality. Can the Authors justify the choice of the DD approach? Is it possible for the Authors to provide simulations that can integrate the results of their study also with a PD approach in comparison with the only DD approach?
Author Response
in file

Reviewer 2 Report
This study is on a topic of relevance and general interest to the readers of the Resources journal.
The manuscript is well written and well organized. The methodology adopted to tackle the problem is solid.
However, the manuscript needs minor improvements. In the enclosed document, there are comments through the manuscript meant to help the authors to improve their work.

Author Response
in file

Reviewer 3 Report
In the paper, the author clearly proposed application of Epanet software in the hydraulic modeling and its use to develop a risk map of lack of water. The paper introduced a supply network distributed water for 200 thousand inhabitants in selected town. The manuscript showns interesting approach to estimations of risk assiociated with insufficient water supply. The research is well designed and implemented, the results obtained are interesting. In my opinion paper can be published after revision.
Minor mistakes were not avoided:
In Table 2, the value of 0.3 failure/km*year was adopted as the limit of a very low risk level. According to the literature on the subject, this is the limit value for main pipelines. If the index exceeds 0.3 failure/km*year, it indicates a poor condition of the pipeline and requires taking appropriate corrective actions. On the other hand, the value of 1.0 indicates a high failure rate, but for water supply connections, which do not play an important role in the water distribution system. It seems that the weights for the parameter P are underestimated.
In this same table, one of the unit is km-1*rok-1? What does it mean “rok”?
In formula number 2, the unit must be completed.
Author Response
in file

Round 2
Reviewer 1 Report
The Authors responded adequately to the observations. The paper can be published in the updated form.